# The Ameliorative Effects of Saikosaponin in Thioacetamide-Induced Liver Injury and Non-Alcoholic Fatty Liver Disease in Mice

**DOI:** 10.3390/ijms222111383

**Published:** 2021-10-21

**Authors:** Geng-Ruei Chang, Wei-Li Lin, Tzu-Chun Lin, Huei-Jyuan Liao, Yu-Wen Lu

**Affiliations:** 1Department of Veterinary Medicine, National Chiayi University, 580 Xinmin Road, Chiayi 60054, Taiwan; lin890090@gmail.com (T.-C.L.); pipi324615@gmail.com (H.-J.L.); 2Bachelor Degree Program in Animal Healthcare, Hungkuang University, 6 Section, 1018 Taiwan Boulevard, Shalu District, Taichung 433304, Taiwan; ivorylily99@gmail.com; 3General Education Center, Chaoyang University of Technology, 168 Jifeng Eastern Road, Taichung 413310, Taiwan; 4Department of Chinese Medicine, Show Chwan Memorial Hospital, 1 Section, 542 Chung-Shan Road, Lukang Township, Changhua 50008, Taiwan; 5Department of Chinese Medicine, Chang Bing Show Chwan Memorial Hospital, 6 Lugong Road, Lukang Township, Changhua 50544, Taiwan

**Keywords:** fatty liver disease, inflammation, liver injury, saikosaponin-d, thioacetamide

## Abstract

Liver disorders are a major health concern. Saikosaponin-d (SSd) is an effective active ingredient extracted from *Bupleurum falcatum*, a traditional Chinese medicinal plant, with anti-inflammatory and antioxidant properties. However, its hepatoprotective properties and underlying mechanisms are unknown. We investigated the effects and underlying mechanisms of SSd treatment for thioacetamide (TAA)-induced liver injury and high-fat-diet (HFD)-induced non-alcoholic fatty liver disease (NAFLD) in male C57BL/6 mice. The SSd group showed significantly higher food intake, body weight, and hepatic antioxidative enzymes (catalase (CAT), glutathione peroxidase (GPx), and superoxide dismutase (SOD)) and lower hepatic cyclooxygenase-2 (COX-2), serum alanine aminotransferase (ALT), aspartate aminotransferase (AST), alkaline phosphatase (ALP), interleukin (IL)-1β, tumor necrosis factor (TNF)-α, and fibroblast growth factor-21 (FGF21) compared with controls, as well as reduced expression of inflammation-related genes (nuclear factor kappa B (*NF-κB*) and inducible nitric oxide synthase (*iNOS*)) messenger RNA (mRNA). In NAFLD mice, SSd reduced serum ALT, AST, triglycerides, fatty acid–binding protein 4 (*FABP4*) and sterol regulatory element–binding protein 1 (*SREBP1*) mRNA, and endoplasmic reticulum (ER)-stress-related proteins (phosphorylated eukaryotic initiation factor 2α subunit (p-eIF2α), activating transcription factor 4 (ATF4), and C/EBP homologous protein (CHOP). SSd has a hepatoprotective effect in liver injury by suppressing inflammatory responses and acting as an antioxidant.

## 1. Introduction

The liver is a vital organ of the body, being involved in bile, protein, and clotting factor production and regulating cholesterol and glycogen metabolism. The liver plays an important role in xenobiotic metabolism, and the portal vein blood supply from the intestinal tract to the liver exposes it to toxins first [1]. It is more prone to injuries compared to any other organ of the body. Liver injuries include histologic lesions (due to cholestasis, inflammation, and necrosis) and others type of injuries (due to cytotoxicity) [2]. Inflammation and wound healing are interrelated processes, since it is inflammatory signals that stimulate immune cells toward injury sites [3]. Hepatocytes are parenchymal cells of the liver, and their apoptosis is reported in liver injury [4]. Non-alcoholic fatty liver disease (NAFLD), or idiopathic steatosis, is seen in patients without any history of alcohol use. The pathophysiology of NAFLD varies from simple steatosis with or without inflammation or fibrosis to non-alcoholic steatohepatitis (NASH), liver cirrhosis, and, ultimately, hepatocellular carcinoma (HCC) [5]. The increased prevalence of NAFLD is a significant public health concern due to increased mortality from liver-related and liver-unrelated causes [6].

Chemical toxins, such as thioacetamide (TAA), carbon tetrachloride (CCl4), acetaminophen, and galactosamine, are used both in vitro and in vivo to induce experimental models of liver injury [7,8]. TAA is widely used for induction of liver injury experimentally. Furthermore, acute TAA application results in inflammation of hepatocytes, while chronic TAA application leads to liver cirrhosis. In addition, TAA’s clinical features are intermittent to chronic human liver disease [9,10]. TAA treatment increases liver transaminases (alanine aminotransferase [ALT] and aspartate aminotransferase [AST]), alkaline phosphatase (ALP), and total bilirubin [11]. In one study, the mean serum ALT, AST, and γ-glutamyltransferase (γ-GT) level in the obesity group significantly increased compared with the normal body mass index (BMI) group [12]. ALT and AST leak into the bloodstream from damaged hepatocytes, which increases their plasma levels. Thus, an increase in plasma AST and ALT activity in the serum is considered a decrease in hepatocytes due to injury [13]. 

TAA and NAFLD stimulate the free-radical production, which causes the deterioration of glutathione and other antioxidant defense mechanisms in hepatocytes, leading to lipid peroxidation [14,15]. The biochemical processing of TAA within the cellular milieu leads to oxidative damage of hepatocytes, which in turn causes degeneration and necrosis in liver tissue [16]. TAA-induced hepatic inflammation, fibrosis, and liver damage in mice are, in terms of clinical features, comparable to chronic liver disease in humans [10,17]. Accordingly, the advantage of using TAA to induce hepatoxicity in animal models is its unique specificity in targeting the liver [17,18]. Animal models play an essential role in demonstrating the pathophysiological mechanisms of NAFLD [19]. NAFLD is induced by feeding C57BL/6 mice (the most used strain for this experimental disease model) a high-fat diet (HFD) [20]. HFD intake causes obesity, hyperinsulinemia, hyperglycemia, hypertension, and liver damage in rats, quite like the phenotype characters reported in humans with NAFLD. An NAFLD mouse model exhibits a pathological spectrum ranging from steatosis to steatohepatitis with fibrosis and later with a probability of developing HCC [21].

In the past, over 100 different triterpenoid saponins have been extracted from *Bupleurum falcatum*, a Chinese herb used for to treat inflammatory and infectious diseases [22]. Saikosaponin-a (SSa), saikosaponin-d (SSd), saikosaponin-c (SSc), and saikosaponin-b2 (SSb2) are the triterpenoids with pharmacological potential [23]. SSd is a main active constituent of *B. falcatum*. It has a steroid-like structure, is reported to be hepatoprotective in animals, and leads to cell cycle arrest and apoptosis in cancer cell lines through control of factors such as p53, nuclear factor kappa B (NF-κB) and Fas/Fas ligands [24,25,26]. SSd reduces the messenger RNA (mRNA) levels of pro-inflammatory cytokines interleukin (*IL*)*-6*, tumor necrosis factor (*TNF*)*-α*, and IL-1β and increases anti-inflammatory cytokine *IL-10* [27]. In addition, SSd decreases ventilator-induced lung injury by reducing oxidative stress and inhibiting inflammatory responses [26]. SSd can be a promising new drug candidate for various chronic age–related diseases, such as Alzheimer’s disease, cardiovascular disease, and obesity [28]. SSd also controls HCC cell proliferation by suppressing the phosphorylated signal transducer and activator of transcription 3 CCAAT/enhancer binding protein beta (p-STAT3/C/EBPβ) signaling pathway and inhibits cyclooxygenase-2 (COX-2) expression [29]. In particular, SSd protects against CCl4- and dimethylnitrosamine-induced liver injury in rats [30]. Thus, SSd has the potential to be developed as an anticancer therapeutic in combination with chemotherapeutics such as oxaliplatin or cisplatin [31,32]. 

Overall, SSd has potential anti-inflammatory, antitumor, neuroregulatory, immunoregulatory, antioxidative, hepatoprotective, and antiviral properties [33]. Although the effects of SSd on CCl4-induced liver injury have been explored, its effects on TAA-induced liver injury and fatty liver are unclear. Therefore, we investigated the mechanisms of SSd treatment against TAA- or obesity-induced liver injury or fatty liver in male C57BL/6J mice. 

## 2. Results

### 2.1. SSd Affects Body Weight and Food Intake in Mice with TAA-Induced Liver Injury

The body weight, daily food intake, daily body weight gain, and daily food efficiency were statistically different in all three groups (control, saline-treated TAA-induced liver injury mice [TAA], and SSd-treated TAA-induced liver injury mice [SSd]; Figure 1). Hepatotoxicity was induced in male C57BL6/J mice by injecting TAA for 6 weeks. After 8 weeks, the SSd group showed upregulated morphological parameters compared with the TAA group. The body weight and food intake of all groups after 8 weeks are depicted in Figure 1a,c, respectively. After 8 weeks, compared with the TAA group, the SSd group had a 1.09-fold higher body weight with statistical significance (Figure 1a), increased body weight gain (weekly) by 1.92-fold (Figure 1b), and 1.41-fold higher food intake (Figure 1c). These results were consistent with significant changes in the daily food efficiency in the SSd group, with a 1.42-fold higher daily food efficiency compared with the TAA group (Figure 1d). However, the daily body weight, food intake, food efficiency, and body weight gain were significantly lower in the SSd group compared with the TAA group.

### 2.2. SSd Affects Liver Weight, Serum ALT, AST, ALP, γ-GT, and Bilirubin in Mice with TAA-Induced Liver Injury

Across the three groups, liver weight and serum ALT, AST, ALP, γ-GT, and bilirubin were statistically different. There was a 43% increase in liver weight in the SSd group (Figure 2a) compared with the TAA group. However, serum ALT, AST, and ALP were downregulated by 17.7% (Figure 2b), 24% (Figure 2c), and 18% (Figure 2d) in the SSd group, respectively, indicating significant downregulation of hepatic function markers compared with the TAA group. In addition, the SSd group demonstrated a 23% and 11% decrease in serum γ-GT (Figure 2e) and serum bilirubin (Figure 2f), respectively. However, compared with the SSd group, serum ALT, AST, ALP, γ-GT, and bilirubin levels and liver weight in the control group were lower.

### 2.3. SSd Reduces Liver Damage in Mice with TAA-Induced Liver Injury

Hematoxylin and eosin (H&E) staining was used to find histopathological changes in all three groups (Figure 3). The histological profile of the control group showed normal hepatocytes with a well-preserved cytoplasm, a prominent nucleus, and a nucleolus, without any inflammation. However, fatty change and necrosis were observed. In the TAA group, extensive injuries, with multifocal areas of hepatocyte necrosis, cell swelling, membrane disruption, and nucleus contraction, were observed. The SSd group showed significantly diminished histological changes (Figure 3a), and the regular hepatocyte structure had much reduced ballooning and tissue degeneration. Necrotic activity was also significantly reduced compared with the TAA group, in addition to the liver damage score using Suzuki scoring (0–4) (Figure 3b) [18,34,35].

### 2.4. SSd Affects Serum IL-1β, TNF-α, Fibroblast Growth Factor-21 (FGF21), and C-Reactive Protein (CRP) Levels in Mice with TAA-Induced Liver Injury

Serum inflammatory cytokines significantly varied among the three groups (Figure 4). Compared with the TAA group, serum IL-1β was 22% lower (Figure 4a) and serum TNF-α was 42% lower in the SSd group (Figure 4b). Serum FGF21 regulates hepatic metabolic pathways to improve steatosis and inflammation [36,37,38]. Compared with the TAA group, serum FGF21 was 2.21-fold higher (Figure 4c) and serum CRP was 24% lower in the SSd group (Figure 4d). Compared with the control group, the SSd group showed a 2.79-fold increase in serum IL-1β, a 1.48-fold increase in serum TNF-α, a 45% decrease in serum FGF21, and a 2.19-fold increase in serum CRP.

### 2.5. SSd Aggregates Hepatic Proteins for COX-2 and the mRNA of Liver Fatty Acid–Binding Protein (L-FABP), Inducible Nitric oxide Synthase (iNOS), and NF-κB

Specific expression of hepatic COX-2 enzyme protects against liver injury by inhibition of apoptotic mechanisms in hepatocytes and promotes cell cycle proliferation and progression [39]. Western blotting analysis showed significantly different COX-2 expression among the three groups (Figure 5a). Compared with the TAA group, COX-2 expression in hepatocytes reduced by 47% (Figure 5b) and hepatic mRNA of *L-FABP* reduced by 54.9% in the SSd group (Figure 5c). Furthermore, *NF-κB* (Figure 5d) and *iNOS* mRNA levels (Figure 5e) in hepatocytes reduced by 33.9% and 67.3% in SSd and TAA groups, respectively. In contrast, compared with the control group, hepatic COX-2 expression was 3.7-fold higher and hepatic *L-FABP*, *NF-κB*, and *iNOS* mRNA levels were 2.19-, 3.97-, and 1.42-fold higher in the SSd group, respectively.

### 2.6. SSd Increases Antioxidant Enzymes and Reduces Reactive Oxygen Species (ROS) in Mice with TAA-Induced Liver Injury

TAA treatment induces liver injury via oxidative damage [18,40], so we investigated whether SSd improves damage to hepatocytes. We found a significant difference in hepatic antioxidant enzymes, such as catalase (CAT), superoxide dismutase (SOD), and glutathione peroxidase (GPx) (Figure 6a–c). The progress of liver injury closely resembles a decrease in hepatic antioxidants; when the activity of these enzymes increases, liver function is enhanced, thereby counteracting any type of hepatotoxicity [18]. The SSd group exhibited a 1.54-, 1.72-, and 1.96- increase in CAT (Figure 6a), GPx (Figure 6b), and SOD (Figure 6c) activity, respectively, compared with the TAA group. ROS overproduction in the liver predominantly increases in liver injury, altering the structure of the liver and consequently causing stark dysfunction of hepatocytes with poor prognosis [41]. The SSd group expressed a significant reduction in hepatic ROS by 23% compared to the TAA group (Figure 6d). CAT, GPx, and SOD reduced by 16%, 54%, and 16%, respectively, in the SSd group compared with the control group. In contrast, hepatic ROS production increased by 2.41-fold in the SSd group compared with the control group.

### 2.7. SSd Affects Body Weight, Fatty Liver, and Body Weight Gain in HFD-Induced NAFLD Mice

Obese mice with HFD-induced NAFLD that were treated with SSd for 8 weeks showed reduced morphometric parameters compared with the HFD-fed control group (Figure 1). Compared to the control group, the SSd group showed significantly reduced body weight by 11% (Figure 7a) and body weight gain by 32.7% (Figure 7b). H&E staining for morphometric analysis of tissues of the control and SSd groups indicated that compared with the control group, the SSd group had a significant decrease in liver fat (Figure 7c), suggesting that SSD decreases hepatic lipids by inhibiting fat deposition in hepatocytes. The SSd group showed a 28% reduction in fatty liver scores compared with the control group (Figure 7d).

### 2.8. SSd Affects ALT, AST, and mRNA Levels of FABP4 and Sterol Regulatory Element–Binding Protein 1 (SREBP1) in HFD-Induced NAFLD Mice

Serum ALT and AST in the SSd group decreased by 24.8% and 32.2% compared with the control group, respectively, indicating hepatic function (Figure 8a,b). SREBP1 and FABP4 are involved in regulating many metabolic pathways, such as those related to type 2 diabetes, atherosclerosis, and hepatic lipid accumulation [42,43]. These genes reportedly have dominant effects in promoting FLD in rodents and humans [44]. In this study, compared with the control group, the SSd group had 21.9% (Figure 4c) and 35.1% (Figure 4d) lower hepatic mRNA levels of *FABP4* and *SREBP1*, respectively. Therefore, SSd reduces gene-associated hepatic steatosis, significantly decreasing AST and ALT levels, and could improve NAFLD symptoms, mediating the mRNA expression of molecular mechanisms to affect hepatic lipid accumulation.

### 2.9. SSd Decreases Serum and Hepatic Triglycerides and the Protein Expression of eIF2α, ATF4, CHOP, and p62 

Compared with the control group, serum (Figure 9a) and hepatic (Figure 9b) triglycerides significantly decreased by 28.5% and 24.8%, respectively, in the SSd group. ER stress causes the onset and development of liver injury [45] and regulates lipogenesis, steatosis, and hepatic autophagic flux [46]. Therefore, we selected genes involved in ER stress, such as phosphorylated eukaryotic initiation factor 2α subunit (p-eIF2α), C/EBP homologous protein (CHOP), activating transcription factor 4 (ATF4), and the autophagy regulator p62, in Western blot analysis (Figure 9c). Remarkably, p-eIF2α/eIF2α, ATF4, and CHOP expression in the SSd group was significantly lower by 14.9%, 81.7%, and 42.4%, respectively) compared with the control group (Figure 9d–f, respectively). The p62 protein appears to play a negative regulatory role in autophagy. The hepatic expression of p62 in the SSd group was significantly lower by 56.5% compared with the control group (Figure 9g). These results indicated that SSd decreases ER stress, increases the gene expression of autophagy in the liver, and improves HFD-induced fatty liver scores.

## 3. Discussion

Liver diseases, including those due to viruses or drugs, are considered serious health problems and require immediate attention in the form of therapies with few side effects [47]. Blood-based biomarkers such as cytokeratin 18, along with various gene polymorphisms (e.g., those in patatin-like phospholipase domain-containing protein 3 and transmembrane 6 superfamily member 2) have been demonstrated to correlate with NAFLD and its severity, and they can potentially be employed in point-of-care testing to risk stratify patients with NAFLD [48,49]. Noninvasive diagnostic tools, in particular the fibrosis-4 index and the NAFLD fibrosis score, are reliable screening tools that can accurately exclude advanced fibrosis and cirrhosis. Moreover, they can be used to dichotomize patients for further workup with readily available clinical and laboratory information [50]. Hepatic inflammation is common in liver injury associated with liver fibrosis, cirrhosis, and HCC. Promising anti-inflammatory molecules can be considered a cornerstone of liver disease treatment [17]. Many studies have demonstrated that SSd has beneficial effects on health via anti-inflammatory, antioxidant, neuroprotective, anticancer, and cardioprotective activities [28,51]. All these properties make SSd a promising option for NAFLD management. In the current study, we successfully established a TAA-induced hepatic injury and HFD-induced NAFLD model to determine the benefits of SSd, a plant-based medicinal bioactive compound. Compared with the TAA group, the SSd group exhibited significantly higher food intake, body weight, CAT, GPx, and SOD, as well as lower COX-2, serum ALT, AST, ALP, IL-1β, TNF-α, and fibroblast growth factor-21 (FGF21). Moreover, the reduced expression of inflammation-related *NF-κB* and *iNOS* mRNA was observed in the SSd group. In the mice with NAFLD, SSd reduced levels of serum ALT, AST, triglycerides, FABP4, and SREBP1 mRNA, as well as the expression of ER-stress-related proteins. SSd protected against liver injury by suppressing inflammatory responses and acting as an antioxidant. Furthermore, it induced the regulation of hepatic inflammatory pathways and other regulatory pathways implicated in NAFLD. 

Natural products are believed to cause fewer side effects and are more compatible with the body’s physiology [52]. The main hurdles in using new natural products include the process of setting up a comprehensive knowledgebase of medicinal plants, purifying components, and studying their mechanisms of action. We developed a model of TAA-induced liver injury and an HFD-induced obese NAFLD model to study the beneficial effects of SSd and successfully demonstrated the regulation of hepatic inflammatory and other related regulatory pathways and fatty liver scores using SSd.

Following TAA treatment for 6 weeks, SSd contributed significantly to a decrease in body weight and body weight gain compared with the control group. A decrease in weight gain occurs when food intake decreases due to a decrease in appetite [53]. This supports the idea that rats with TAA-induced liver injury undergo changes in the tryptophan levels in the brain, resulting in a decrease in food intake related to anorexia [54]. Anorexia, discoloration of stool, nausea, yellow blood-like urine, jaundice, and heartburn are common characteristics of liver disorders [55]. The degree of anorexia was weak in these studies and was directly proportional to the energy status of the animal’s body during the time of induction of inflammation [18,56]. Rats in which inflammation was induced lost 6% of their body weight, resulting in the most severe anorexia [57]. TAA treatment changes the hepatic histology and significantly upregulates plasma ALT and AST levels [58,59]. In this study, SSd prevented liver damage and ameliorated the histological damage to the liver, with a significant decrease in ALT and AST levels compared with the TAA group. Taken together, SSd acts as a potential agent to improve liver-damage-related anorexia and increase daily food efficiency in mice with liver injury.

An increase in plasma liver enzyme levels, such as ALP, bilirubin, and γ-GT, indicates liver injury [60]. We also found a considerable increase in serum ALT, AST, ALP, bilirubin, and γ-GT [18,61]. Zhu et al. reported that SSa significantly decreases serum AST and ALT levels increased by d-galactosamine-induced liver injury and lipopolysaccharides by increasing liver X receptor [62]. In addition, fruits of the plant Vitex doniana possess antioxidant properties and with chrysin, saikosaponin, and ellagitanin extracts prevent acetaminophen-mediated increase in serum ALP and bilirubin [63]. In Figure 2e of this study, plasma γ-GT levels in the SSd group decreased, which was similar to SS improving the hepatic fibrosis response in mice with CCl4-induced injury [64]. Furthermore, SSd treatment led to significant recovery of hepatocytes, as demonstrated by the reduced plasma levels of hepatic enzymes. These findings helped us validate the hepatoprotective activity of SSd. 

Besides biochemical results, similar effects were observed in histopathological findings in mice treated with SSd. While some studies have reported an increase in liver weight due to accumulation of collagen and extracellular matrix (ECM) protein in animals after TAA treatment for over 6 weeks [65,66], we found, using Sirius Red staining, that mice with liver injury induced by 6-week treatment with TAA do not develop liver fibrosis (Appendix A). However, inconsistencies in our finding with other studies are relatively due to the duration of TAA treatment. Concerning the research question about the effect of liver damage and fibrogenesis, many studies have suggested the hepatoprotective role of medicinal plants in the TAA injury model [18]. In our disease model, SSd treatment significantly attenuated liver injury by SSd’s anti-inflammatory effects, as shown by decreased serum ALT, AST, ALP, bilirubin, and γ-GT and improved tissue pathologic changes, in parallel with lower Suzuki histological scores after 56 days of SSd treatment compared with the TAA group, which showed huge necrosis and hepatocellular swelling. SSd led to significant differences in discriminating between stages, as evaluated according to the Desmet–Scheuer classification of inflammation (Appendix A). In chronic hepatic injury experiments performed in combination with different studies, SSd, like other triterpenoid saponin chemicals, such as SSa, SSb2, and SSc, is responsible for restoration of hepatic physiology [33,62]. Furthermore, SSd plays a promising role in preventing hepatic injury.

The liver plays a potential role in inflammation by its ability to synthesize a number of proteins that participate in the systemic inflammatory response [67]. Therefore, we investigated inflammatory markers, such as IL-1β, TNF-α, FGF21, and CRP. The activation of NF-κB signaling can also be caused by IL-1β, upregulation of pro-inflammatory cytokines, and hepatocellular damage [68]. Mice with IL-1β deficiency have reduced liver fibrosis and diet-induced inflammation [69]. TNF-α, a pro-inflammatory cytokine, induces the initial phase of the inflammatory process and has extensively represented the systemic inflammation response and liver injury in inflammatory diseases [70]. Studies have reported reduced CCl4-induced liver fibrosis in TNF-receptor-knockout mice [71,72,73]. Moreover, plasma AST, ALT, and γ-GT levels decrease with increasing FGF21 [18]. A beneficial effect was observed when treatment with recombinant FGF21 produced significant protective effects on overall liver function [74]. CRP is a prime marker of inflammation, and increased CRP levels show that chronic ethanol administration is responsible for severe hepatic endothelial injury [75,76]. Our results also demonstrated that IL-1β, TNF-α, and CRP levels in the SSd group were lower compared with the TAA group. In addition, the IHC staining showed that the levels of inflammatory cytokines, such as IL-1β (Appendix A), were lower in the SSd group. In contrast, FGF21 was increased after SSd treatment, indicating that SSd can reduce inflammation and the severity of liver injury. Therefore, SSd demonstrates significant hepatoprotective effects against TAA-induced liver injury by regulating the inflammatory process.

COX-2 expression has been related to the early phases of different chronic liver diseases and to the induction of HCC [77]. L-FABP can be used for diagnosis of acute hepatitis, chronic hepatitis, and cirrhosis [78]. Upregulated serum L-FABP levels are related to the degree of hepatic fibrosis and inflammation, which indicates serum L-FABP can be a non-invasive marker to assess the severity of fibrosis and inflammation in NAFLD patients [79]. NF-κB plays an essential role in the regulation of inflammatory signaling pathways in the liver and contributes to liver homeostasis and wound-healing processes by genetic inactivation of different NF-κB signaling components [80,81]. iNOS are reported to be involved in excessive production of pro-inflammatory mediators [82]. Chronic liver diseases begin with an inflammatory phase, followed by fibrosis and continuous oxidative stress. In this state, iNOS production increases, causing large amounts of nitric oxide production [83]. In Western blotting, SSd treatment reduced COX-2 expression. Moreover, the gene expression of *L-FABP*, *NF-κB*, and *iNOS* was lower in the SSd group compared with the TAA group. Therefore, SSd can reduce inflammation and liver-damage-related factors, thereby reducing the effect of TAA-induced liver injury.

Oxidative stress is a major factor of the progression of liver damage, and ROS induce hepatic inflammation, necrosis, and cholestasis [84]. Antioxidant enzymes, including CAT, GPx, and SOD, offer protection against the harmful effects of ROS [85]. GPx is a family of enzymes that constitute the main antioxidant defense system in mammals [86]. GPx and SOD levels are regularly downregulated with an accompanied increase in ALT and AST levels [87]. CAT is present in peroxisomes, where it decomposes two hydrogen peroxide (H_2_O_2_) molecules into two H_2_O molecules and O_2_ [88]. As the concentration of H_2_O_2_ increases, CAT shows a greater contribution to H_2_O_2_ degradation [89]. Serum CAT activity moderately increases in fatty liver and acute alcoholic hepatitis [90]. SSd decreased oxidative stress and ROS content by increasing GPx, SOD, and CAT activity. Similarly, SSa, as a free-radical scavenger, restores decreased SOD, CAT, and GPx activities in rats with nephritis [91]. Therefore, in TAA-induced liver injury, SSd treatment can reduce ROS to prevent further damage to hepatocytes.

Mice were given SSd for 8 weeks after an HFD diet for 10 weeks; the diet contributed to obesity development. Weight increases when the fat pad mass increases because of an increase in the number of new adipocytes from precursor cells or because of enlarged adipocytes due to fat storage [42,92]. Weight gain is related to increased fat cell differentiation or fat pad weight caused by fat cell hypertrophy [92,93]. For this reason, the liver histology of the SSd group was healthier and the fatty liver score was lower compared to the control group. The SSd group also showed improved glucose intolerance, as demonstrated by the area under the curve (AUC) for 120 min after glucose injection (Appendix A). The result of Oil Red O staining also indicated that the SSd group had less Oil Red O accumulation (Appendix A). Additionally, the increase in body fat may have been due to lower energy expenditure, indicated by the mRNA expression in brown adipose tissue of uncoupling protein 1 (*UCP1*; Appendix A). UCP1 expression occurs in the inner membrane of brown adipocyte mitochondria to create heat through the uncoupling of oxidative phosphorylation, and thermogenesis induction is controlled by adrenaline. Hypothalamus feeding centers are linked to this thermogenic system for body temperature control, which enables the regulation of body weight and may well represent an efficient dual function related to brown fat thermogenesis [94]. Therefore, the SSd group could improve fatty problem by increasing energy expenditure.

Fat accumulation in the liver induces FLD, which is strongly associated with obesity [42,95]. The SSd group had lower fatty liver scores (as revealed in histopathology) and lower AST and ALT levels (liver enzymes and liver injury markers). Hepatocellular permeability increases when liver injury is induced, and consequently, AST and ALT are released from intracellular spaces into plasma [96]. In addition, *SREBP1* and *FABP4* mRNA expression are associated with the expression of genes participating in lipid storage, hepatic steatosis, hepatic lipogenesis, and NAFLD pathogenesis [97]. Lower expression of *SREBP1* and *FABP4* mRNA in the liver is observed after long-term SSd administration and improved liver injury due to significant hepatic lipid infiltration. 

Obesity is highly associated with increased triglycerides [98], and accumulated triglycerides in the hepatocytes’ cytoplasm are the hallmark of NAFLD [99]. For this reason, the SSd group showed increased serum and hepatic triglyceride levels compared with the control group. ER stress responses in adipose tissue are a key factor in the aggravation of obesity-related problems, and PERK-eIF2α-ATF4 signaling is most important [100,101]. Activation of the PERK-eIF2α-ATF4 signaling pathway by antipsychotic drugs increases intracellular lipid accumulation via activation of SREBP-1c and SREBP-2 in hepatocytes [102]. ATF4 is important in the regulation of lipid metabolism. ATF4 overexpression induces triglyceride accumulation in HepG2 cells, indicating the impact of ATF4 on lipid metabolism in the liver [103]. CHOP is an ER-stress-induced transcription factor that is a significant mediator of apoptosis, which is a key mechanism for disease progression in patients with NAFLD [104,105]. CHOP deficiency suppresses β-cell loss in murine models of ER-stress-mediated diabetes and type 2 diabetes [106]. Thus, SSd treatment could improve the symptoms of fatty liver by reducing ER stress and apoptosis. In addition, one factor in the argument that autophagy plays a role in NAFLD pathophysiology is that autophagic function of hepatic cells is impaired under several conditions that predispose to NASH [107]. Activating autophagy via increased AMPK phosphorylation, decreased mTOR, and increased LC3B expression [108]. Autophagy can prevent NAFLD and AFLD progression through enhanced lipid catabolism and decreased hepatic steatosis [109]. Our study observed the expression of p62 that actually appears to play a negative regulatory role in autophagy, and p62 accumulation is observed in the liver of ob/ob mice, and its aggregation is correlated with serum ALT activity and inflammatory activity by the NAFLD score [110]. Also, hepatocyte lipid accumulation is associated with reduced autophagic function in hepatic cells [111]. Thus, SSd could ameliorate steatosis, inflammation, and enhanced lipid catabolism via autophagy.

Together, the study findings demonstrated that SSd has therapeutic hepatoprotective activity mediated via inhibition of inflammatory pathways, inhibition of ER stress, and upregulation of antioxidant enzymes and autophagy. Hence, we propose the use of SSd as a novel promising hepatoprotective agent for liver disorders.

## 4. Materials and Methods

### 4.1. Animal Experiments

Male C57BL/6J mice aged five-weeks and weighing 18–20 g were purchased from the Education Research Resource, National Laboratory Animal Center, Taiwan. The mice were housed in cages at 22 °C ± 1 °C and 55% ± 5% humidity under a 12:12 h light:dark cycle and fed a standard diet (3.3 kcal/g of metabolizable energy; diet 5008; PMI Nutrition International, St. Louis, MO, USA) for 4 weeks ad libitum. Finally, we selected 40 mice, each weighing ~25 g. 

Housing of all mice and experimentation were performed according to the Guidelines for the Care and Use of Laboratory Animals as well as in accordance with the Taiwan government’s recommendations. The protocol was approved by the National Chiayi University’s Institutional Animal Care and Use Committee (archive no. 108017).

#### 4.1.1. Liver Injury

The mice were divided into three groups (n = 8 per group): control, TAA, and SSd (TAA + SSd) groups. An intraperitoneal injection of 100 mg/kg of TAA (Sigma-Aldrich, St. Louis, MO, USA) [67,112] was administered to the TAA and SSd groups for 6 weeks. Then, the SSd group was administered SSd (2 mg/kg; Sigma-Aldrich) in saline by oral gavage once a day for 8 weeks. SSd is a potential molecule for treating liver fibrosis and is a liver protectant in a mouse model [113,114,115]. In our preliminary studies, serum ALT and AST levels after 1 mg/kg of SSd did not differ significantly from those in the TAA group (Appendix A). Hence, we used SSd at a dose of 2 mg/kg. All mice were weighed weekly. At the end of 8 weeks, the mice were euthanized, and their livers and sera were collected. We analyzed inflammatory cytokines, liver function antioxidant enzymes, and molecular proteins, and performed histological examination, RNA extraction, and quantitative polymerase chain reaction (qPCR) analysis.

#### 4.1.2. NAFLD

The remaining 16 mice were fed an HFD (5.16 kcal/g of metabolizable energy; diet 58Y1, modified lab w/31.66% lard, 1.12 μg/g chromium; PMI Nutrition International) ad libitum for 10 additional weeks, which is larger than the typical 4-week duration adopted in obesity-related studies [43]. After obesity was induced, some of the then 19-week-old mice were divided into two groups (n = 8 per group), control and SSd. The average body weight of the control and SSd groups was 32.14 ± 1.02 and 31.58 ± 0.87 g, respectively, with no significant differences. The SSd group was administered SSd (2 mg/kg) in saline by oral gavage daily for 8 weeks. The control group was given only saline by oral gavage. All mice were weighed weekly. At the end of 8 weeks, the mice were euthanized, and their livers and sera were collected to determine liver enzymes and conduct a molecular protein analysis.

### 4.2. Body Weight and Food Intake

We performed weekly measurements of food intake and body weight throughout the study period. The leftover food in each cage dispenser and food spillage were used to measure food intake. 

### 4.3. Liver Function Enzyme Tests

An automated chemistry analyzer (Catalyst One Chemistry Analyzer; IDEXX Laboratories, Westbrook, ME, USA) was used to quantify serum ALT, AST, ALP, bilirubin, globulin, and γ-GT levels according to the manufacturer’s instructions, with a coefficient of velocity of <2%.

### 4.4. Liver Histopathological Evaluation

We measured liver weights and determined their percentage in relation to the total body weight in liver injury experiments. Isolated liver tissues were embedded in paraffin, treated with formalin, and subjected to histological analysis. To quantify liver damage, 3-μm-thick liver tissue sections were stained with H&E. A four-point scale (Suzuki scoring of 0–4) was used to classify liver injuries. The degree of liver damage was assessed as follows: 0 = normal; 1 = development of a sinusoidal congestion space; 2–3 = the presence and/or severity of sinusoidal congestion and cytoplasmic vacuolization; and 4 = necrosis of parenchymal cells and hemorrhage [29,30].

In NAFLD experiments, harvested livers were weight in grams. H&E staining was used to study fat infiltration into the liver. The scoring was as follows: 0 = no fat infiltration; 1 = <5% fat infiltration; 2 = 5–25% fat infiltration; 3 = 25–50% fat infiltration; and 4 = >50% fat infiltration [19,38,93]. A high-resolution digital microscope was used to perform imaging (Moticam 2300; Motic Instruments, Richmond, BC, Canada), and Motic Images Plus version 2.0 software (Motic Instruments, Richmond, BC, Canada) was used to determine the histology and morphometry of liver.

### 4.5. Determination of Serum Inflammatory Cytokines, FGF21, and CRP

Plasma TNF-α and IL-1β levels were determined using mouse TNF-α and IL-1β enzyme-linked immunosorbent assay (ELISA) kits (Invitrogen, Carlsbad, CA, USA) according to the manufacturer’s instructions. Serum mouse FGF-21 ELISA kits were obtained from Zgenebio Biotech (Taipei, Taiwan). Serum CRP levels were measured using a mouse CRP quantitative ELISA kit (Abcam, Cambridge, UK).

### 4.6. Western Blotting Assay

The mice were anesthetized using an intraperitoneal injection of urethane. Their livers were quickly removed, coarsely minced, and immediately homogenized prior to Western blotting [42] using anti-COX-2 (Sigma-Aldrich), p-eIF2α (Cell Signaling Technologym, Beverly, MA, USA), eIF2α (Sigma-Aldrich), ATF4 (Cell Signaling Technologies), CHOP (Sigma-Aldrich), and p62 (Sigma-Aldrich) anti-actin antibodies. We used enhanced chemiluminescence reagents (Thermo Fisher Scientific, Rockford, MA, USA) to produce immunoreactive signals and UVP ChemStudio (Analytik Jena, Upland, CA, USA) to detect these signals. Protein expression and phosphorylation experiments were quantified using ImageJ software (National Institutes of Health, Bethesda, MA, USA).

### 4.7. RNA Extraction and Real-Time qPCR

The mRNA levels of *L-FABP*, *NF-**κB*, *iNOS, FABP4 and SREBP1* in the liver were determined using the CFX Connect Quantitative Real-Time PCR System (Bio-Rad Laboratories, Hercules, CA, USA). Briefly, total RNA was isolated from liver samples using TRI Reagent (Sigma-Aldrich), and its concentration was assessed based on absorbance at 260–280 and 230–260 nm using a Qubit fluorometer (Invitrogen). Next, we performed real-time qPCR using complementary DNA (cDNA) and the iTaq universal SYBR Green Supermix (Bio-Rad Laboratories) according to the manufacturer’s instructions. The qPCR protocol was as follows: 95 °C for 5 min, then 45 cycles at 95 °C for 15 s, followed by 60 °C for 25 s. The sequence primers were as follows:

*L-FABP:* forward 5′-GCAGAGCCAGGAGAACTTTG-3′ and reverse 5′-GGGTCCATAGGTGATGGTGAG-3′

*NF-**κB*: forward 5′-ATGGCTTCTATGAGGCTGAG-3′ and reverse 5′-GTTGTTGTTGGTCTGGATGC-3′

*iNOS*: forward 5′-TCTTGGGTCTCCGCTTCTCGTC-3′ and reverse 5′-TGGCTGGTACATGGGCAC AGAG-3′

*FABP4:* forward 5′-GATGAAATCACCGCAGACGACA-3′ and reverse 5′-ATTGTGGTCGACTTTCCATCCC-3′

*SREBP1:* forward, 5′-CGGAAGCTGTCGGGGTAG-3′ and reverse 5′-GTTGTTGATGAGCTGGAGCA-3′ 

Each target gene expression level was calculated in relation to Actb levels and expressed using the 2^−ΔΔCt^ method.

### 4.8. Evaluation of Hepatic CAT, GPx, and SOD

The functional activity of the antioxidant system was analyzed to evaluate hepatic CAT, GPx, and SOD activities. The liver was perfused with ice-cold saline (0.9% sodium chloride) and homogenized in chilled 1.17% potassium chloride [94]. The homogenates were centrifuged at 800× *g* for 5 min at 4 °C. The supernatant was recentrifuged at 10,500× *g* for 20 min t 4 °C to finally obtain the postmitochondrial supernatant, which was used to measure CAT, GPx, and SOD activities. These enzymes were quantified using a commercially available colorimetrical kit (#K773-100 for CAT, #K762-100 for GPx, and #K335-100 for SOD; BioVision, Milpitas, CA, USA).

### 4.9. Statistical Analysis 

All data are expressed as means ± standard error of the mean (SEM). In the comparison of liver injury, all experimental data, except for pathological findings, were analyzed by one-way analysis of variance (ANOVA), followed by Duncan’s multiple-range test. The data of NAFLD and pathological examination of liver injury scores were analyzed using the *t*-test. A *p*-value of <0.05 was considered statistically significant.

## 5. Conclusions

The study demonstrated that SSd is effective in preventing TAA-induced liver injury in mice. The protective effect of SSd in TAA-induced liver injury would possibly contribute to modulation of its antioxidant and free-radical-scavenging effects. The underlying mechanism is increased CAT, GPx, and SOD expression; decreased ROS production; decreased inflammatory conditions, as revealed by the decrease in serum TNF-α, IL-1β, FGF21, and CRP levels; inhibition of hepatic COX-2 expression; and decrease in *L-FABP*, *NF-κB*, and *iNOS* mRNA levels, which ultimately prevent the progression of hepatic damage in mice. Furthermore, the constant intake of SSd has good effects on obesity and HFD-induced NAFLD in mice. Parallel alterations in weight gain, food efficiency, serum triglyceride levels, serum AST and ALT levels, fatty liver scores, and ER stress are also indicated. Moreover, the results evidence the mechanism underlying SSd-induced improvement of fatty liver by decreasing ER-stress-related protein expression. Hence, our findings not only validate the use of SSd as a hepatoprotective agent but also support its future exploration as a therapeutic drug for liver disorders. 

## Figures and Tables

**Figure 1 ijms-22-11383-f001:**
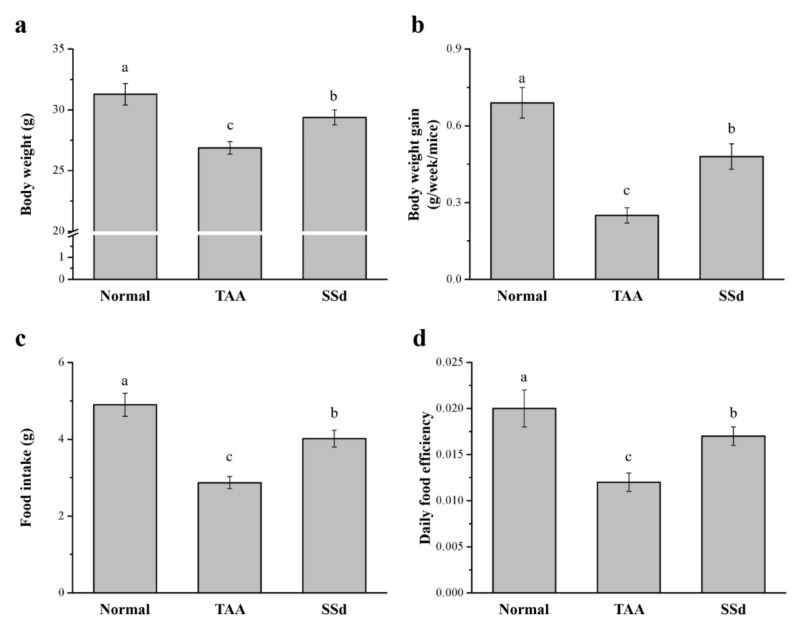
Changes in (**a**) body weight, (**b**) weekly body weight gain, (**c**) food intake, and (**d**) daily food efficiency in the control, TAA, and SSd groups over 8 weeks. All data are reported as means ± SD (n = 8). ^a–c^ Data with different letters above the columns were significantly different on one-way ANOVA, and Duncan’s test at *p* < 0.05 was used to compare the means of the three groups. TAA, thioacetamide; SSd, saikosaponin-d; SD, standard deviation; ANOVA, analysis of variance.

**Figure 2 ijms-22-11383-f002:**
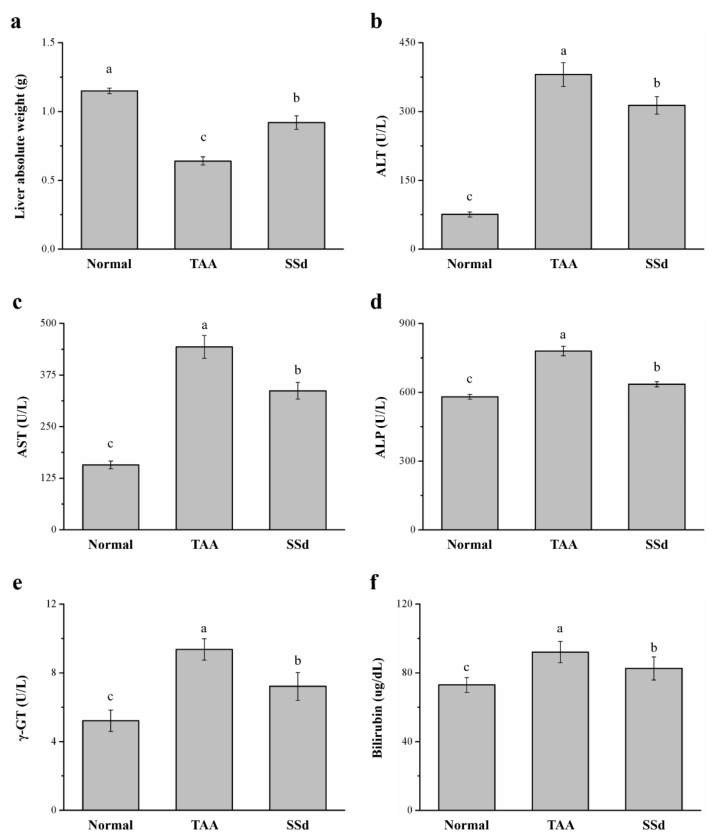
Changes in (**a**) liver weight and serum (**b**) ALT, (**c**) AST, (**d**) ALP, (**e**) γ-GT, and (**f**) bilirubin levels in the control, TAA, and SSd groups over 8 weeks. All data are reported as means ± SD (*n* = 8). ^a–c^ Data with unlike letters above the columns were significantly different on one-way ANOVA, and Duncan’s test at *p* < 0.05 was used to compare the means of the three groups. ALT, alanine aminotransferase; AST, aspartate aminotransferase; ALP, alkaline phosphatase; γ-GT, γ-glutamyltransferase.

**Figure 3 ijms-22-11383-f003:**
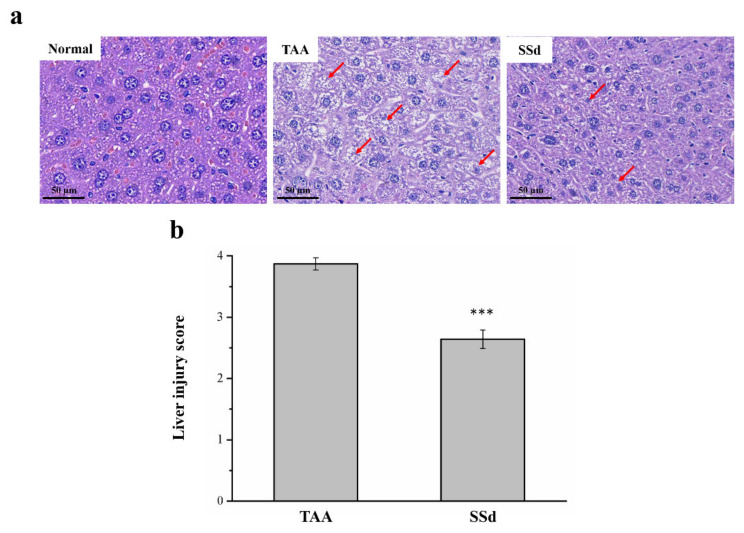
Changes in (**a**) liver histology by H&E staining (magnification 200×) and (**b**) scoring index of liver injury in the control, TAA, and SSd groups over 8 weeks. Anormal hepatocytes are indicated by red arrows. All data for are reported as means ± SD (n = 8). *** *p* < 0.001 was considered statistically significant. H&E, hematoxylin and eosin.

**Figure 4 ijms-22-11383-f004:**
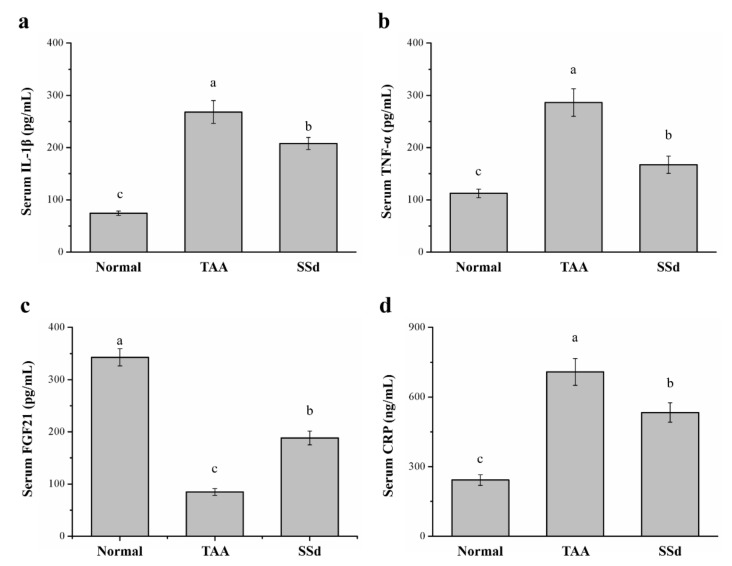
Changes in serum (**a**) IL-1β, (**b**) TNF-α, (**c**) FGF21, and (**d**) CRP levels in the control, TAA, and SSd groups over 8 weeks. All data are reported as means ± SD (n = 8). ^a–c^ Data with different letters above the columns were significantly different on one-way ANOVA, and Duncan’s test at *p* < 0.05 was used to compare the means of the three groups. IL-1β, interleukin 1 beta; TNF-α, tumor necrosis factor alpha; FGF21, fibroblast growth factor-21; CRP, C-reactive protein.

**Figure 5 ijms-22-11383-f005:**
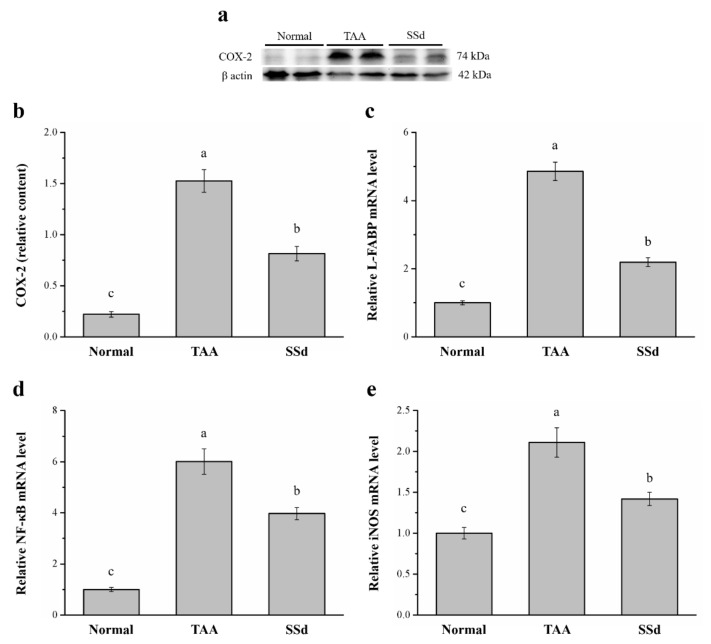
(**a**) Representative Western blot of liver extracts for COX-2 expression in the control, TAA, and SSd groups. (**b**) COX-2 expression and changes in (**c**) *L-FABP* (**d**) *NF-κB*, and (**e**) *iNOS* mRNA levels in hepatocytes in the control, TAA, and SSd groups over 8 weeks. All data are reported as means ± SEMs (n = 8). ^a–c^ Data with different letters above the columns were significantly different on one-way ANOVA, and Duncan’s test at *p* < 0.05 was used to compare the means of the three groups. COX-2, cyclooxygenase-2; L-FABP, liver fatty acid–binding protein; NF-κB, nuclear factor kappa B; iNOS, inducible nitric oxide synthase; mRNA, messenger RNA; SEM, standard error of the mean.

**Figure 6 ijms-22-11383-f006:**
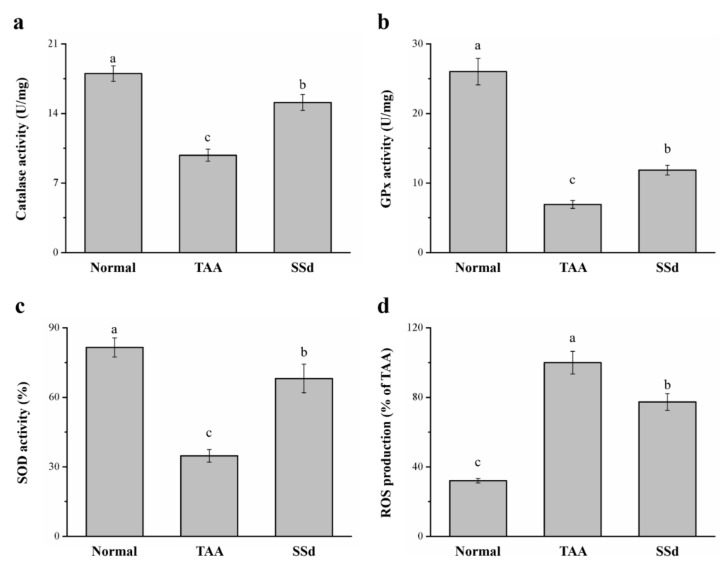
Changes in hepatic (**a**) CAT, (**b**) GPx, and (**c**) SOD activity and (**d**) ROS levels in the control, TAA, and SSd groups over 8 weeks. All data are reported as means ± SD (n = 8). ^a–c^ Data with different letters above the columns were significantly different on one-way ANOVA, and Duncan’s test at *p* < 0.05 was used to compare the means of the three groups. GPx, glutathione peroxidase; CAT, catalase; SOD, superoxide dismutase; ROS, reactive oxygen species.

**Figure 7 ijms-22-11383-f007:**
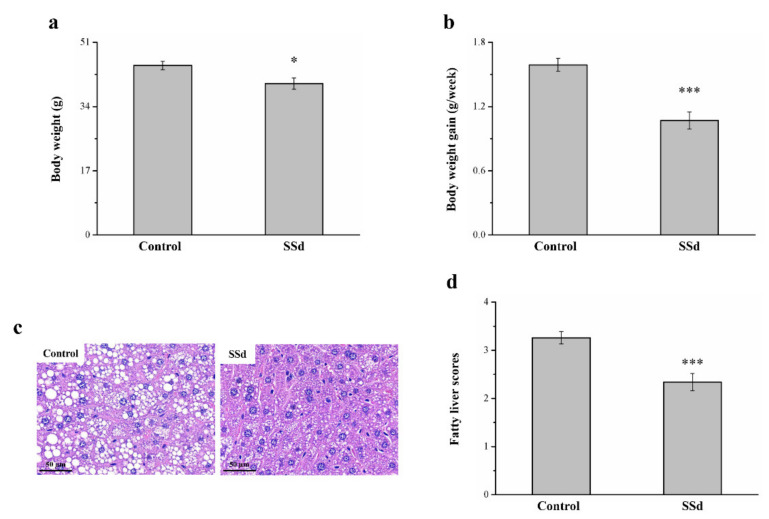
Changes in (**a**) body weight and (**b**) weekly body weight gain in the HFD-fed control and SSd groups. (**c**) H&E-stained liver tissue sections of control and SSd groups and (**d**) fatty liver scores of control and SSd groups over 8 weeks. All data are reported as means ± SD (n = 8). * *p* < 0.05; *** *p* < 0.001.

**Figure 8 ijms-22-11383-f008:**
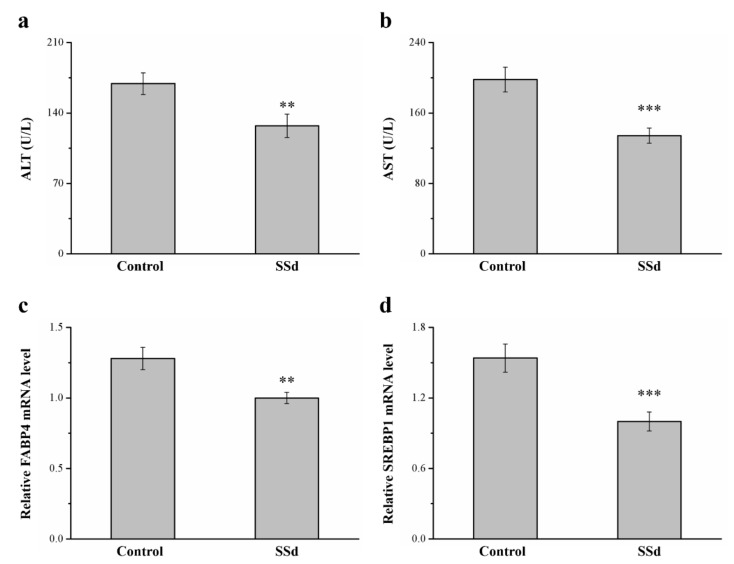
Effects of SSd on (**a**) serum ALT, (**b**) serum AST, and hepatic mRNA levels of (**c**) *FABP4* and (**d**) *SREBP1* measured in the control and SSd groups. All data are reported as means ± SD (n = 8). ** *p* < 0.01; *** *p* < 0.001. FABP4, fatty acid–binding protein 4; SREBP1, sterol regulatory element–binding protein 1.

**Figure 9 ijms-22-11383-f009:**
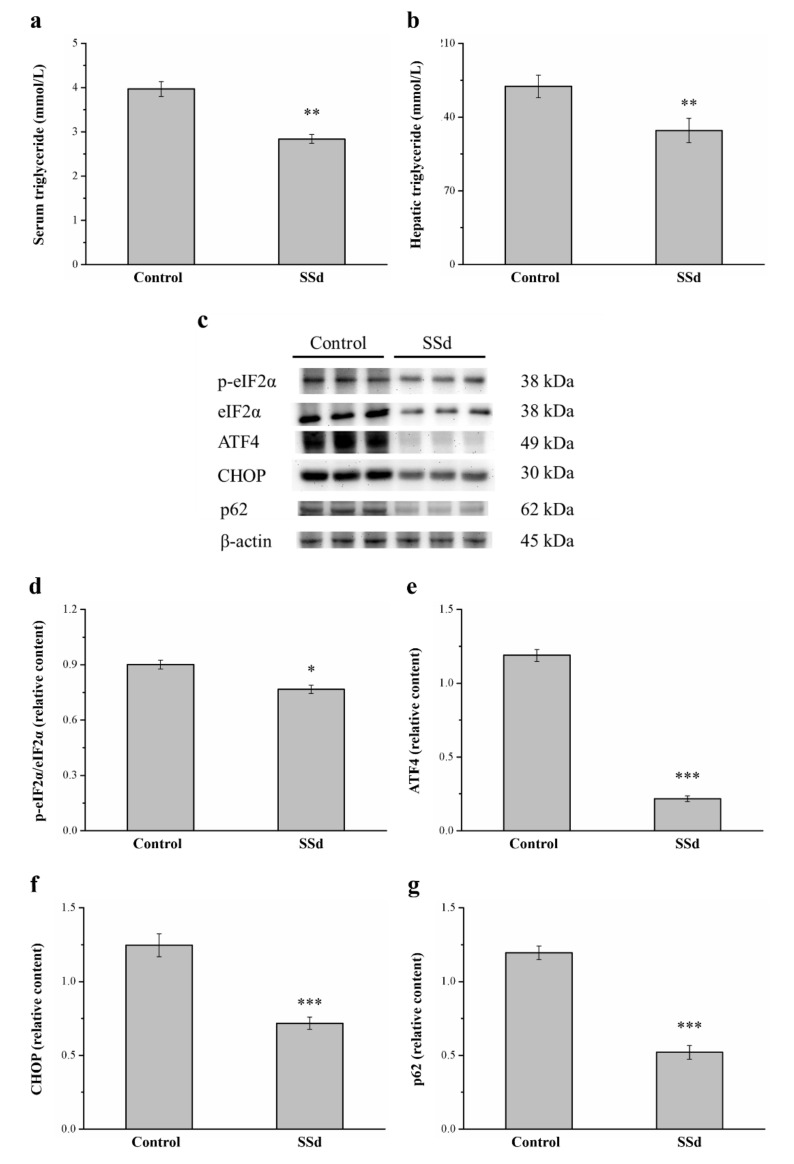
Effects of SSd treatment on (**a**) serum and (**b**) hepatic triglycerides. (**c**) Representative image showing blots of liver extracts. The expression of (**d**) p-eIF2α/eIF2α, (**e**) ATF4, (**f**) CHOP, and (**g**) p62 was measured. All data are reported as means ± SD (n = 8). * *p* < 0.05; ** *p* < 0.01; *** *p* < 0.001. p-eIF2α, phosphorylated eukaryotic initiation factor 2α subunit; ATF4, activating transcription factor 4; CHOP, C/EBP homologous protein.

## Data Availability

The data presented in this study are available on request from the corresponding author.

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
