# Peer review of "The Ameliorative Effects of Saikosaponin in Thioacetamide-Induced Liver Injury and Non-Alcoholic Fatty Liver Disease in Mice"

_ijms, 2021, doi:10.3390/ijms222111383_

Round 1

Reviewer 1 Report

In the present study, Geng-Ruei Chang et al. explore the ameliorative effects of saikosaponin in experimental liver disease. The study is straight forward and provides further promising insights into potential therapeutical approaches. However, there are some issues that need too be solved before acceptance:

  1. A more comprehensive analysis of histological examinations should be provided, including grading according to Desmet and Scheuer with assessment of inflammatory lesions.
  2. There should be confirmation of mRNA data shown for FABP4 and SREBP1. 
  3. Genetic nomenclature should be italic (mRNA, C57BL6). 
  4. Supplementary Figure 5 states SSd at 1 mg/kg, the main figures 2 mg/kg. Please clarify. 

Reviewer 2 Report

The mansucript co-authored by Chang et al. describes the The Ameliorative Effects of Saikosaponin in Thioacetamide-Induced Liver Injury and Non-alcoholic Fatty Liver Disease in Mice. Authors investigated the effects and underlying mechanisms of saikosaponin-d (SSd) treatment for thioacetamide (TAA)-induced liver injury and high-fat-diet (HFD)-induced non-alcoholic fatty liver disease (NAFLD) in male C57BL/6 mice. Interestingly the SSd group demonstrated effectively higher food intake, body weight, and hepatic antioxidative enzymes (catalase [CAT], glutathione peroxidase [GPx], and superoxide dismutase [SOD]) and lower hepatic cyclooxygenase-2 (COX-2), serum alanine aminotransferase (ALT), aspartate aminotransferase (AST), alkaline phosphatase (ALP), interleukin (IL)-1β, tumor necrosis factor (TNF)-α, and fibroblast growth factor-21 (FGF21) compared with controls, as well as reduced expression of inflammation-related genes (nuclear factor kappa B [NF-κB] and inducible nitric oxide synthase [iNOS]) messenger RNA (mRNA). The paper is well written and in my opinion would be of interest to readers involved in Ameliorative drug design and evaluation. I consider the manuscript as suitable for publication after minor revision.

[1] The plant name « Bupleurum falcatum » should be italic in all manuscript.

[2] The authors mentioned in Material method section about the source of saikosaponin-d (SSd).

Reviewer 3 Report

Reviewer comments and suggestions

The author investigated the effects and mechanisms of Saikosaponin-d (SSd) treatment for thioacetamide (TAA)-induced liver injury and high-fat-diet (HFD)-induced non-alcoholic fatty liver disease (NAFLD) in mice. The SSd group showed significantly elevated hepatic antioxidative enzymes and decreased liver marker levels. Moreover, in NAFLD mice, SSd decreased serum ALT, AST, triglycerides, fatty acid-binding protein 4 (FABP4) and sterol regulatory element-binding protein 1 (SREBP1) mRNA, and endoplasmic reticulum (ER)-stress-related proteins (phosphorylated eukaryotic initiation factor 2α subunit [p-eIF2α], etc. Therefore, the author suggested that SSd may have a hepatoprotective effect in liver injury by targeting inflammatory responses and acting as an antioxidant.

Decision: Minor comments

Below are the comments for this paper to be incorporated in the revised version of the manuscript. 

  1. Line 19, the medicinal plant name should be in italic line 81-82
  2. Line 42-43 two times cholestasis was used
  3. These lines do not match with the previous line and also forwards lines, better to delete it “Abnormal mitochondrial functioning because of apoptotic signaling cascades can lead to a decline in cellular adenosine triphosphate and result in a necrotic morphology”
  4. In the introduction, the authors can mention about some useful biomarkers to diagnosis NAFLD such as PMC4999188, PMC5437499, Nat Rev Gastroenterol Hepatol 2018 Aug;15(8):461-478. https://doi.org/10.1016/j.metabol.2020.154259, https://doi.org/10.1002/edm2.127, the author can mention the above information in the discussion para line 279
  5. Line 72 please explore the references here
  6. Line 97 what does it indicate combined therapy
  7. Please delete these references and lines 98-99, also line 105-107
  8. For fig 3 better include astar to indicate types of cells
  9. Figure 7 need to change under the histogram for better clarity “obese SSd groups”
  10. It would be better to include the study observation in the first para of the discussion
  11. Line 301-304 Please check the lines
  12. Line 321 Mention figure or table number
  13. Please add more references in line number 346
  14. Line 352-354 missing points, please correct it
  15. Line 413 is the information was correct 
  16. In the end the author mention about autophagy “it needs to be explored more in the paper as the author did not discuss as compared to the other points”
  17. Please check reference 12

Round 2

Reviewer 1 Report

The authors addressed all my comments, I recommend acceptance of the manuscript in its current version.